# Ellagic Acid as a Tool to Limit the Diabetes Burden: Updated Evidence

**DOI:** 10.3390/antiox9121226

**Published:** 2020-12-03

**Authors:** Antonio J. Amor, Carmen Gómez-Guerrero, Emilio Ortega, Aleix Sala-Vila, Iolanda Lázaro

**Affiliations:** 1Endocrinology and Nutrition Service, Hospital Clínic de Barcelona, 08036 Barcelona, Spain; EORTEGA1@clinic.cat; 2Centro de Investigación Biomédica en Red de la Fisiopatología de la Obesidad y Nutrición (CIBEROBN), Instituto de Salud Carlos III (ISCIII), 28029 Madrid, Spain; 3Renal and Vascular Inflammation Lab, Health Research Institute-Fundación Jiménez Díaz (IIS-FJD), Autonoma University of Madrid (UAM), 28040 Madrid, Spain; cgomez@fjd.es; 4Centro de Investigación Biomédica en Red de Diabetes y Enfermedades Metabólicas Asociadas (CIBERDEM), Instituto de Salud Carlos III (ISCIII), 28029 Madrid, Spain; 5Institut d’Investigacions Biomèdiques August Pi Sunyer (IDIBAPS), 08036 Barcelona, Spain; 6IMIM—Hospital del Mar Medical Research Institute, 08003 Barcelona, Spain; asala3@imim.es; 7Barcelonaβeta Brain Research Center, Pasqual Maragall Foundation, 08005 Barcelona, Spain

**Keywords:** ellagic acid, type 2 diabetes, glucose metabolism, oxidative stress, inflammation, diabetic complications

## Abstract

Oxidative stress contributes not only to the pathogenesis of type 2 diabetes (T2D) but also to diabetic vascular complications. It follows that antioxidants might contribute to limiting the diabetes burden. In this review we focus on ellagic acid (EA), a compound that can be obtained upon intestinal hydrolysis of dietary ellagitannins, a family of polyphenols naturally found in several fruits and seeds. There is increasing research on cardiometabolic effects of ellagitannins, EA, and urolithins (EA metabolites). We updated research conducted on these compounds and (I) glucose metabolism; (II) inflammation, oxidation, and glycation; and (III) diabetic complications. We included studies testing EA in isolation, extracts or preparations enriched in EA, or EA-rich foods (mostly pomegranate juice). Animal research on the topic, entirely conducted in murine models, mostly reported glucose-lowering, antioxidant, anti-inflammatory, and anti-glycation effects, along with prevention of micro- and macrovascular diabetic complications. Clinical research is incipient and mostly involved non-randomized and low-powered studies, which confirmed the antioxidant and anti-inflammatory properties of EA-rich foods, but without conclusive results on glucose control. Overall, EA-related compounds might be potential agents to limit the diabetes burden, but well-designed human randomized controlled trials are needed to fill the existing gap between experimental and clinical research.

## 1. Introduction

Type 2 diabetes (T2D) is a growing epidemic, with an estimated prevalence of 463 million people nowadays worldwide and is thought to increase to 700 million by 2045 [1]. Micro- and macrovascular complications are the main cause of morbidity and mortality in diabetic patients [2]. Despite the increasing number of pharmacological options [3], dietary intervention still remains as the cornerstone of the prevention and treatment of T2D [4]. Oxidative stress is a long-known contributor to the T2D pathogenesis [5] and diabetic vascular complications [6,7]. Therefore, dietary intake of antioxidants might contribute to limiting the burden of diabetes through multiple ways.

There is compelling evidence of the health benefits of dietary polyphenols, a type of plant-derived antioxidants [8]. Among polyphenols, tannins and ellagitannins, in particular, are believed to protect against diabetic vascular complications [9]. Ellagitannins are found in walnuts [10] and pomegranates, as well as at lower concentrations in several berries [11,12], some beverages, such as brandy and oak-aged red wine [13], and some medicinal plants [14]. Upon hydrolysis, ellagitannins release ellagic acid (EA), which is further metabolized in colon to urolithins [15,16,17].

In addition to the well-known effect in inhibiting cancer-cell proliferation [10,18,19,20], further effects have been recently described for EA and its derived metabolites [12]. The link between dietary EA and chronic diseases is attracting increasing interest. In this review are update findings from experimental and clinical studies on EA and the burden of diabetes, delving into three aspects: First, the mechanisms underlying the effect of EA on glucose metabolism; second, how dietary EA might limit diabetic-driven tissue damage via its antioxidant, anti-inflammatory, and anti-glycation properties; and third, whether dietary EA protects against diabetic complications. We include studies involving EA given in isolation, extracts or preparations enriched in EA, and EA-rich foods, mostly pomegranate juice. Walnuts were left out on purpose because of their accumulation of salutary phytochemicals other than EA that act synergistically, blurring the exact contribution of EA to a specific effect. To identify all relevant literature published on the topic, MEDLINE and SCIENCE CITATION INDEX searches were performed, ending February 2020. The keywords used were (ellagitannin* OR ellagic acid* OR urolithin*) AND (diabetes OR glucose OR retinopathy OR nephropathy OR kidney disease OR neuropathy OR cardiovascular OR diabetes complications* OR inflammation* OR glycosylation*). In addition, references cited in published original and review articles were examined until no further study was identified.

## 2. Ellagic Acid and Glucose Control

### 2.1. In Vitro Studies

Some in vitro models focused on the anti-hyperglycemic effects of EA. In HepG2 hepatocytes under high glucose, EA (15, 30 μM) increased glucose consumption, being the effect of the highest dose akin to the well-known insulin-sensitizer drug metformin (150 μM). EA upregulated insulin receptor substrate-1 (IRS1), Akt and extracellular signal-regulated kinase (ERK) phosphorylation under insulin stimulation [21]. Similarly, EA stimulated glucose-induced insulin secretion in mice isolated islets, yet not reaching the effect of the secretagogue drug tolbutamide [22]. Stimulation of insulin secretion was also observed in rat pancreatic islets treated with EA [23]. Many in vitro models reported a dose-dependent effect of EA and its derivatives in inhibiting α-glucosidase activity. The studies included EA-rich foods (pomegranate juice [24]), extracts from EA-rich plants, such as pomegranate [25,26,27,28] and Mongolian oak cups [29], as well as EA [24,26,27,29] and its gut-derivate urolithin A [24]. A minor inhibitory activity in other carbohydrates hydrolases (such as α-amylase) has also been described [26,29]. Dipeptidyl peptidase IV (DPP-IV) has also been studied as a target of EA. Pomegranate juice, EA, and urolithin A displayed a strong effect in murine 3T3-L1 adipocytes, along with inhibition of pancreatic lipase [30]. The in silico DPP-IV inhibition observed after the addition of the extract of arjun tree (Terminalia arjuna) was similar than that observed after the challenge with sitagliptin and vildagliptin, two DPP-IV inhibitors currently used for T2D [30]. Finally, in differentiated murine 3T3-L1 adipocytes, the addition of pomegranate fruit extract or EA significantly suppressed resistin secretion (an adipokine associated with dyslipidemia and insulin resistance) via promoting its degradation at protein level [31].

### 2.2. Animal Studies

#### 2.2.1. Non-Diabetic Murine Models

In mice under a high-fat diet, 12-week intraperitoneal administration of urolithin A (20 μg/day) significantly improved systemic insulin sensitivity, attenuated liver steatosis, and reduced adipocyte hypertrophy and macrophage infiltration into the adipose tissue [32]. Fifteen-day dietary pomegranate peel extract (200 mg/kg/day) in non-diabetic Wistar albino rats lowered serum glucose levels, while insulin remained unchanged [33]. In female ddY mice subjected to bilateral ovariectomy (a model of insulin resistance), 12-week dietary pomegranate fruit extract (30 mg/kg/day) significantly reduced serum resistin levels without changing serum glucose [31]. Similarly, in high-fat and high-sucrose diet-induced obesity mice (but without diabetes), extracts of pomegranate flower and peel (250 mg/kg/day) did not improve fasting glucose levels after 2 or 4 weeks, nor glucose after oral glucose tolerance test (OGTT) or insulin tolerance test after 2 weeks of intervention [34]. The same results were found in high-fat diet-induced obese Balb/c mice, in which 4-week supplementation with pomegranate peel extract (6 mg/day) neither improved glycemia nor glucose tolerance [35]. A study characterizing an extract of the aerial parts of pea plant (with a high content of polyphenols, including EA) reported that oral treatment with 200 mg/kg 30 min before OGTT displayed an anti-hyperglycemic effect in non-diabetic Swiss-albino mice [36]. Finally, 8-week dietary inclusion of 7% of raspberry pomace in non-diabetic Wistar albino rats reduced plasma glucose levels, particularly if the raspberry pomace was subjected to fine grinding, a process which increased the polyphenol concentration. The authors related this effect to that observed in the gastrointestinal tract (inhibition of α-glucosidase and β-glucuronidase), along with the plasmatic elevation of the fibroblast growth factor-19 [37].

#### 2.2.2. Streptozotocin-Induced Diabetic Murine Models

Streptozotocin (STZ) causes pancreatic beta (β)-cell destruction and, subsequently, compromises insulin production, leading to an insulin-dependent diabetes mellitus. Twelve-week old, neonatal STZ-induced diabetic rats were given a single dose of EA (25, 50, or 100 mg/kg) 60 min prior to OGTT. EA decreased peak glucose in a dose-dependent manner. The highest dose induced a reduction similar to that of standard drug glibenclamide [22]. Two studies assessed the effects of longer exposures. In the first one, 3-week EA (30 mg/kg/day) improved fasting plasma glucose in a similar way to moderate doses of pioglitazone, a well-established anti-hyperglycemic agent. The combination of EA (10 mg/kg/day) and pioglitazone (10 mg/kg/day) acted synergistically [38]. In the second study, 8-week EA (50 mg/kg/day) decreased fasting plasma glucose to a similar extent than subcutaneous Neutral Protamine Hagedorn (NPH) insulin (6 UI/day) [39].

STZ-induced diabetic murine models have repeatedly been used to test pomegranate extracts. Oral administration of a single dose of pomegranate seed extract (600 mg/kg) lowered by 52% the blood glucose at 12 h, inducing a higher effect than a dose of 200 mg/kg of chlorpropamide [40]. Longer exposures have also been tested. Pomegranate peel extract (5, 15, 25, and 100 mg/kg administered every 2 days for 10 days) significantly reduced fasting blood glucose in a dose-dependent manner, while increasing insulin levels [41]. Similar results were observed after 4-week treatment with extracts of pomegranate leaf or fruit peel (100 and 200 mg/kg/day), although not reaching that observed with glibenclamide [42]. Pomegranate flowers aqueous extract (250 and 500 mg/kg for 3 weeks) also reduced fasting blood glucose and improved lipid parameters in STZ-induced diabetic rats [43]. Finally, 4-week oral administration of pomegranate peel extract alone (400 mg/kg/day) or in combination with black bean peel extract (200 mg/kg/day each) ameliorated hyperglycemia and limited the loss of pancreatic mass secondary to STZ administration [44].

Other extracts from EA-rich medicinal plants or vegetables have also been studied. In a high-fat diet and STZ-induced diabetic rats, 8-week administration of an extract of Chinese olive (*Canarium album* L., 50 and 150 mg/kg/day) limited body weight gain and hepatic steatosis, and reduced glucose in parallel with improving of insulin signaling pathway (decrease in phosphorylated IRS1 and up-regulation of phosphorylated Akt protein expression) [45]. Other studies tested extracts from amla (*Emblica officinalis*, 250 and 500 mg/kg/day) [22] or arjun tree (*Terminalia arjuna*, 500 mg/kg/day) [30] in STZ-induced diabetic rats. The first study, which involved a 4-week intervention, reported on decrease in plasma glucose in a dose-dependent manner. This was believed to be mediated by changes in pancreatic β-cell functionality [22]. In the second one, a 5-week supplementation with the extract significantly restored both glucose and hemoglobin A1c (HbA1c) levels to a similar extent than the antidiabetic drug vildagliptin [30].

Finally, two studies assessed the effect of pomegranate juice (1 mL/day for 3 weeks) on glucose parameters in STZ- and nicotinamide-induced diabetic Sprague-Dawley rats [46,47]. Glucose remained unchanged [46] or significantly decreased [47] compared to rats fed control diet. However, both studies reported benefits in the β-cell function, by increasing plasma insulin levels [47] and increasing average size and number of islets of Langerhans in the pancreas [46]. Finally, pomegranate juice (100 and 300 mg/kg/day) was used in a similar diabetic model but with a longer exposure (6 weeks). Animals fed with pomegranate juice, particularly those at the highest dose, showed a significant decrease in plasma glucose, along with an increase in the insulin levels and a decrease in Homeostatic Model Assessment for Insulin Resistance (HOMA-IR) [48].

#### 2.2.3. Alloxan-Induced Diabetic Murine Models

Alloxan-induced diabetes is another form of insulin-dependent diabetes mellitus, because of a partial degradation β-cells in pancreatic islets. In this experimental model, pomegranate peel extracts were orally administered in three independent studies at doses ranging from 75 to 200 mg/kg/day for 15 days [49], 6 weeks [50], and 8 weeks [51]. All studies consistently reported on a decrease in glucose levels in parallel with an increase in insulin secretion [49,50,51]. One-week administration of an ethanolic extract of pomegranate leaves (500 mg/kg/day) decreased blood glucose and increased tissue glycogen content (liver, skeletal and cardiac muscles) [52]. A study explored the glycemic effects after acute and 3-week administration of an aqueous extract of pomegranate arils at different doses (100, 200, and 350 mg/kg/day). The authors found that both treatments lowered glycemia, as assessed either by fasting blood glucose or OGTT, and increased insulin levels through a modulation of the phosphoinositide 3-kinase (PI3K)/Akt pathway [53]. Finally, a similar effect in glucose metabolism was observed after the administration of pomegranate rind extract (or its spray dried biopolymeric dispersions) [54].

Extracts from EA-rich plants have also been studied. A single dose (250 or 500 mg/kg) of an ethanolic extract from leaves of maire (*Crataegus azarolus* var. *eu-azarolus*, a spiny tree rich in phenolic components, EA being one of the most abundant) significantly decreased plasma glucose after OGTT [55]. A repeated administration (30 days) also induced an hypolipidemic effect [55]. Fifteen-day supplementation with an ethanolic extract from Mongolian oak cups (800 mg/kg/day) improved glucose levels and lipid parameters [56].

Finally, a study supplemented alloxan-induced diabetic Sprague-Dawley rats with either walnut leaf, coriander leaf or pomegranate seed (60 g/kg/day, each) for 15 days. Only supplementation with walnut leaf significantly reduced blood glucose, while increasing density of islets in pancreatic tissue, percentage of β-cells, and islet size [57].

#### 2.2.4. Other Diabetic Murine Models

In an experimental model of insulin resistance/T2D (albino rats fed with a high-fat and high-fructose diet for 2 months), 2-week EA supplementation (10 mg/kg/day) improved glucose/insulin balance in serum, while enhancing insulin signaling, autophosphorylation, adiponectin receptors, glucose transporters, and apoptotic markers in glucose-sensitive tissues (i.e., liver, pancreas, adipose tissue, and brain) [58]. In contrast, in KK-Ay mice fed high-fat diet (a model for T2D possessing insulin resistance), dietary supplementation with EA (0.1%) for 68 days did not affect serum glucose or insulin, although it reduced serum resistin and improved serum lipid profile and hepatic steatosis [59]. Four-week dietary supplementation with EA (50 mg/kg/day) reduced fasting blood glucose, insulin resistance, and liver steatosis in adult female Goto Kakizaki rats (a non-obese spontaneous model of T2D) [60].

Finally, 2-week inclusion of the methanolic extract of pomegranate flower (500 mg/kg/day) lowered plasma glucose levels in non-fasted Zucker diabetic fatty rats (a genetic model of obesity and T2D), whereas it had little effect in fasted animals. In addition, the extract inhibited the increase of plasma glucose levels after a challenge with sucrose load [25].

### 2.3. Human Studies

Most of the human studies have used pomegranate juice as a source of EA (Table 1). Three studies evaluated the effects of the acute administration of this juice on glycemic parameters [61,62,63]. In the only study including participants with T2D, blood samples were collected after a 12-h fast, then 1 and 3 h after administration of 1.5 mL of pomegranate juice per kg body weight. A significant decrease in plasma glucose was found at 3 h, especially in those at lower initial fasting plasma glucose levels. An improvement both in insulin resistance (HOMA-IR) and in β-cell function (HOMA %B) underlined this hypoglycemic effect [61]. Similarly, in healthy adults, concomitant consumption of bread and pomegranate juice (but not a pomegranate-based supplement) significantly attenuated the area under the curve for bread-derived glucose and peak blood glucose [62]. In contrast, no effects were found in a study including pediatric patients with some features of metabolic syndrome but without T2D [63]. Longer exposures have been tested in heterogeneous populations (Table 1). Two non-randomized studies conducted in T2D participants found mixed results for fasting plasma glucose [64,65]. The bulk of randomized clinical trials on the topic [63,66,67,68,69,70,71,72,73,74,75,76] reported on neutral results, except for a study conducted in healthy patients, in which 4-week consumption of pomegranate juice lowered fasting plasma insulin and HOMA-IR compared to a control with equivalent amount of carbohydrates [69].

Pomegranate extracts were also used in several randomized clinical trials, but none of them included individuals with T2D (Table 1) [62,77,78]. No effects on blood glucose were observed after an acute dose in healthy subjects [62]. Studies involving longer exposures yielded mixed results. Compared to participants receiving placebo, no effect was found after supplementation in overweight subjects with increased waist size [77], while significant reductions in fasting plasma insulin and HOMA-IR were observed in obese and overweight subjects [78]. Finally, no hypoglycemic effects were found when performing a OGTT concomitantly or 10 h after consuming a pomegranate/grape pomace dietary supplement [79].

## 3. Anti-Inflammatory, Antioxidant, and Anti-Glycation Properties of Ellagic Acid

### 3.1. Ellagic Acid and Inflammation

Mounting evidence on human and animal models indicates the crucial role of inflammation in the onset and progression of diabetes and its related vascular complications [80,81]. In brief, diabetes is characterized by a low-grade systemic inflammation, with a permanent activation of the main inflammatory pathways (such as nuclear factor-κB, (NF-κB)) and the production of pro-inflammatory cytokines and chemokines (i.e., tumor necrosis factor-alpha (TNF-α), interleukin (IL)-6 or IL-1β), mainly by macrophages. Different anti-inflammatory pharmacological agents have been tested to reduce inflammation in diabetes. However, non-pharmacological or lifestyle interventions are gaining interest to prevent or control diabetes and its complications.

#### 3.1.1. In Vitro Studies

Two in vitro studies have assessed the anti-inflammatory properties of EA. In the first one, EA (20 µM) decreased the expression of cyclooxygenase-2 (COX-2) secondary to the incubation of human aortic endothelial cells with glucose 30 mM [82]. In the second one, an extract of pomegranate flowers (10, 25, 50, 100 μg/mL) decreased the expression of COX-2 in a dose-dependent manner and reduced the synthesis of nitric oxide (NO), prostaglandin E2, IL-6, IL-1β, and TNF-α in lipopolysaccharide-stimulated RAW 264.7 macrophages [83].

#### 3.1.2. Animal Studies

In rat models treated with high-fat and high-fructose diet [58] or STZ [39], EA decreased both circulating and cerebral TNF-α and IL-6 [39,58], as well as increased the anti-inflammatory cytokine IL-10 in hippocampus and cerebral cortex [39].

Similar results were observed after treating mice under a high-fat and high-sucrose diet with either pomegranate flower extract or pomegranate seed oil for 4 weeks [34]. Furthermore, 4-week supplementation with pomegranate peel extracts (6 mg/day) counteracted the expression of COX-2, IL-1β and IL-6 in both colon and visceral adipose tissue (but not in liver) in high-fat-induced hypercholesterolemia and inflammatory disorders [35]. Eight-week treatment with Chinese olive (*Canarium album* L.) extract (50 and 150 mg/kg/day) significantly reduced hepatic IL-6 and circulating and hepatic TNF-α in high-fat diet and STZ-induced diabetic rats [45].

Finally, in STZ-induced diabetic Sprague-Dawley rats, a 3-week supplementation with pomegranate juice (1 mL/day) significantly decreased systemic TNF-α, IL-6 and NF-κB [46]. Similar results regarding TNF-α were observed after a 6-week supplementation (100 and 300 mg/kg/day) in STZ and nicotinamide-induced diabetic Wistar albino rats [48].

#### 3.1.3. Human Studies

Two studies explored the effect of supplementation with pomegranate, as a source of EA. In the first one, a non-randomized trial conducted in 17 subjects, 4-week daily administration of 2 capsules of pomegranate polyphenols (753 mg polyphenols/capsule) did not change high-sensitivity C-reactive protein (hs-CRP) [84]. The second study was a randomized double-blind clinical trial, with a larger sample size, conducted in subjects with overweight or obesity. After a 30-day intervention, participants receiving dietary supplementation of pomegranate fruit extract showed significantly lower circulating hs-CRP and IL-6 levels than those allocated into the placebo group [78].

The anti-inflammatory effect of pomegranate juice have been explored in both non-randomized [65] and randomized clinical trials [63,70,71,73,76,85], using a broad range of juice doses and time of exposure and including participants with either T2D [65,70,85], metabolic syndrome [63,76] or arterial hypertension [71]. Consumption of pomegranate juice reduced serum hs-CRP [70,76], IL-6 [63,65] and leukocyte adhesion molecules [63,71,85], even inducing anti-inflammatory changes in peripheral blood mononuclear cells (decrease in NF-κB and increase in sirtuin-1) [85]. Inflammation-related parameters only remained unchanged in a study conducted in healthy endurance-based athletes [73].

### 3.2. Ellagic Acid and Oxidative Stress

Oxidative stress plays a pivotal role in cellular injury from hyperglycemia [81]. High glucose levels can stimulate reactive oxygen species (ROS) production. Although a certain amount of ROS is necessary for the normal metabolic and signaling cellular processes, cells have developed powerful antioxidant systems to scavenge the surplus of ROS (or at least transform them into less reactive products) to keep redox homeostasis. Oxidative stress results when cellular antioxidant enzymatic machinery is unable to cope with an excessive production of ROS. In diabetes, oxidative stress can damage DNA (which can be assessed by 8-hydroxy-2′-deoxyguanosine, [8-OHdG] and 8-oxo-7,8-dihydro-2′-deoxyguanosine), induce lipid peroxidation (assessed by thiobarbituric acid-reactive substances (TBARS) and malondialdehyde (MDA) levels), and induce protein covalent modifications [81].

#### 3.2.1. In Vitro Studies

EA (1500 µM) reduced ROS production and lipid peroxidation (MDA) in cultured rat pancreatic islets [23]. Similar results were found in high glucose-induced T2D HepG2 cells, with a further increase in superoxide dismutase (SOD) activity [21]. Finally, in intact rat aortas and human aortic endothelial cells under high-glucose conditions, EA (20 µM) decreased ROS production, in parallel with the decreased expression of NOX4 gene [82]. Regarding to EA-rich extracts, pomegranate peel extract reduced the MDA production of erythrocytes incubated with H_2_O_2_ [33]. Furthermore, pomegranate peel extract [86], as well as extracts from several EA-rich plants, such as amla [22], pea plant [36], and maire [55], increased the radical scavenging activity in the 1,1-diphenyl-2-picryl-hydrazyl (DPPH) assay. Finally, pomegranate juice sugar fraction reduced the macrophage peroxide levels in non-diabetic and STZ-induced diabetic mice in a dose-dependent manner, while increasing paraoxonase-2 (PON-2) activity [87].

#### 3.2.2. Animal Studies

EA has been tested in a single study, conducted in high-fat and high-fructose diet-induced diabetic rat model [58]. A 2-week treatment (10 mg/kg) decreased circulating levels of MDA, while increasing those of glutathione (GSH, a potent antioxidant) and the activity of the antioxidant enzymes catalase (CAT) and SOD. Rats on EA displayed lower levels of several caspases and myeloperoxidase in liver, pancreas, and adipose tissue compared to those on chow [58].

Seven studies tested extracts from several parts of pomegranate (leaf, aril, peel, or flower) in non-diabetic [33] and diabetic murine models [42,43,44,49,50,54,88]. Studies repeatedly reported on decreased circulating and tissue (hepatic, pancreatic, cardiac, or renal) lipid peroxidation markers, mostly TBARS [33,42,43,49,50]. Further antioxidant properties were also documented, including decreased ROS [88], increased total antioxidant capacity and GSH levels [43,44,49], and increased (either circulating or tissue-specific) antioxidant enzymes (SOD, CAT, glutathione-S-transferase (GST), glutathione peroxidase (GPx) or glutathione reductase (GR)) [42,43,50,54]. Similar results concerning lipid peroxidation and the decrease in oxidative stress were reported after dietary inclusion of extracts from EA-rich vegetables, including amla [22], Chinese olive [45], and Mongolian oak cups [56].

Finally, the antioxidant effect of pomegranate juice has been assessed in STZ-induced diabetic murine model in four different studies [46,47,48,87]. Again, a decrease in lipid peroxidation (MDA [47,48] or macrophage peroxide levels [87]) and oxidative stress (increased radical scavenging activity in DPPH assay [46,87], GSH content or total antioxidant capacity [47,48,87] or antioxidant enzyme activity [47,48]) was observed. Interestingly, the antioxidant effect of pomegranate juice resembled to the long-known antioxidant ascorbic acid [46].

#### 3.2.3. Human Studies

Several forms of pomegranate have been used as a source of EA to test its antioxidant properties in human studies. Pomegranate fruit extract has been tested in three studies. In the only randomized, double-blind, and placebo-controlled trial, 30-day supplementation with pomegranate fruit extract (1000 mg/day) reduced MDA levels in overweight/obese participants [78]. Three other non-randomized studies tested 4-week supplementation with pomegranate extracts, reporting decreased lipid peroxidation [77,84] and further antioxidant properties, including decreased overall serum oxidative stress and increased high-density lipoprotein (HDL)-associated PON-1 binding and activity, with a greater effect in men than in women [89].

Two randomized clinical trials explored the effects of pomegranate juice in T2D population. Both studies reported an increase in total antioxidant capacity [75,90], along with decreased MDA formation [90], decreased oxidized-low-density lipoprotein (LDL) particles, and increased PON-1 activity [75]. Similarly, in a randomized, double-blind, controlled placebo trial conducted in healthy endurance-based athletes, compared to placebo, 3-week consumption of pomegranate juice (200 mL/day) blunted lipid peroxidation (formation of circulating carbonyls and MDA) secondary to exercise [73]. Finally, pomegranate juice was also tested in four non-randomized studies performed in T2D [64,65,89,91] and healthy [92] subjects. Overall, a decrease in the lipid peroxidation (reduction in circulating MDA levels [64,91,92] and in intracellular peroxides in monocyte-derived macrophages [91]) was consistently observed. These trials also reported a decrease in oxidative stress, including increased PON-1 activity [64,91] and decreased macrophage oxidized-LDL uptake [91].

### 3.3. Ellagic Acid and Glycation

Another way for hyperglycemia to cause injury is by fostering advanced glycation-end products (AGEs) [93]. AGEs result from the non-enzymatic addition of reducing sugars (glycation) to amine-containing molecules (protein, lipids, or nucleotides). Carboxymethyllysine (CML) and pentosidine are two well-characterized AGE compounds, the levels of which serve as AGEs markers. AGEs induce the expression of receptors for advanced glycation end-products (RAGE). The interaction AGE-RAGE promotes ROS generation, activates NF-κB, and therefore increases the expression of pro-inflammatory molecules. AGEs represent a major contributor for the development of diabetic complications. Its deleterious effect is dual: by the accumulation-induced direct tissue damage; and indirectly by the exacerbation of oxidative stress and inflammation. Therefore, counteracting AGEs emerge as an interesting therapeutic approach.

#### 3.3.1. In Vitro Studies

EA has been repeatedly found to inhibit glycation, by means of a reduction in AGEs formation [29,94,95] and by a decrease in fructose-mediated glycation of albumin [96]. Anti-glycation properties of EA (but not of EA-derived metabolites urolithin A and B) were more potent than those observed after treatment with aminoguanidine, a well-known glycation inhibitor [94]. Similarly, both pomegranate fruit [94] and peel [95] extracts showed a dose-dependent reduction in AGEs formation, having the former a higher effect than aminoguanidine [94]. Furthermore, Mongolian oak cups extracts decreased AGEs generation [29]. Finally, in a study testing the effect of commonly consumed juices (pomegranate, cranberry, black cherry, pineapple, apple, and Concord grape) on the fructose-mediated glycation of albumin, while pomegranate juice reduced glycation by 98%, other juices only inhibited glycation by 20% [96].

#### 3.3.2. Animal Studies

A single study has assessed the effects of a pomegranate fruit extract on glycation. In high-fat and high-sucrose-induced diabetic rats, 8-week treatment with pomegranate fruit extract (1.5% *v*/*v*) reduced serum AGEs, glycoalbumin, and HbA1c in a similar manner after treatment with aminoguanidine [95]. Results concerning HbA1c were further confirmed in a study testing EA in isolation, wherein STZ-induced diabetic rats were fed with chow or chow supplemented with either 0.2% or 2% of EA. After a 12-week period, inclusion of EA reduced the increase of HbA1c in a dose-dependent manner [97].

#### 3.3.3. Human Studies

In a randomized, double-blind, placebo-controlled trial, 44 participants with T2D were randomly assigned to pomegranate juice (250 mL/day) or placebo for 12 weeks. At the end of the study, no between-group differences were observed concerning the AGEs markers carboxymethyllysine and pentosidine [90].

## 4. Ellagic Acid and Complications of Diabetes

### 4.1. Microvascular Complications

#### 4.1.1. Diabetic Kidney Disease

Diabetic kidney disease (DKD) is present in approximately 40% of patients with T2D, being the leading cause of chronic kidney disease worldwide [98]. It accounts from minor albuminuria to advanced kidney dysfunction with need of renal replacement therapy [98]. Although some glucose-lowering drugs have recently shown promising results in the prevention and management of this chronic complication [99], common antioxidant agents have also been associated with an improvement in early renal damage [100]. Thus, antioxidants could be used as an additive treatment for the management of this prevalent complication associated with pharmacological intervention.

Data on EA and DKD are limited to experimental research (Table 2). Animal studies were conducted in STZ-induced diabetes [41,97,101,102,103,104] or in high-fat diet combined with low-dose STZ [105]. Four studies tested EA at different doses and with different times of exposure [97,101,104,105]. These studies consistently reported on strongly nephroprotective properties, including amelioration of renal histopathology alterations [104,105]; decreased renal inflammation [101,104,105] secondary to NF-κB inhibition [105]; decreased oxidative stress/lipid peroxidation and increased enzymatic antioxidant activity [105]; reduced fibrosis markers [97,105]; decreased renal AGEs accumulation [97,101]; and inhibition of the polyol pathway in the kidney [101]. All these effects translated into the improvement of the common laboratory kidney function parameters, namely decreased creatinine, or blood urea nitrogen (BUN), increased creatinine clearance, and reduced proteinuria. Interestingly, one study found renal effects of EA akin to those observed with the nephroprotective drug irbesartan [104]. A similar protection was observed when using pomegranate leave extract [103], pomegranate peel extract-stabilized gold nanoparticles [41], or pomegranate juice [102] as exposure.

In vitro experiments further confirmed the EA anti-inflammatory properties on cultured rat proximal tubular epithelial cells challenged to pro-inflammatory stimuli. One study found that EA decreased the high-glucose-induced NF-κB activation and reduced the levels of the inflammatory cytokines IL-1β, IL-6, and TNF-α [105]. In line with this, a second study found that EA inhibited the TLR4/NF-κB inflammatory pathway activated by the administration of lipopolysaccharide [104].

#### 4.1.2. Retinopathy

Diabetic retinopathy (DR) is present in roughly 40% of the individuals with diabetes. Overall, DR is the most frequent cause of blindness among adults aged 20–74 years in developed countries [106]. Although glucose control is strongly associated with the prognosis of this complication, some recent studies have linked some dietary factors (independently of glucose metabolism) with the protection against DR [107,108].

Data on EA and DR is limited to experimental research (Table 2). Two studies conducted in STZ-induced diabetic rats reported protective effects of EA. In the first one, compared to rats fed chow, those fed with EA for 12 weeks showed reductions in the retinal AGE carboxymethyllysine along with activation of receptor of AGEs [109]. EA also reduced pro-apoptotic, neovascularization, and gliosis markers, increased retinal thickness, and improved electroretinogram abnormalities [109]. In the second one, 10-week dietary supplementation with pomegranate juice reduced retinal 8-OHdG and MDA, along with higher GSH and GSH-glutathione peroxidase activities [110].

### 4.2. Macrovascular Complications

Cardiovascular disease is the leading cause of mortality among T2D patients [111]. In fact, this population segment has consistently shown an accelerated atherosclerosis compared to their non-diabetic counterparts [112,113]. Furthermore, and additionally to the increased risk of coronary heart disease, T2D individuals also show a specific heart phenotype (so-called diabetic cardiomyopathy), which increases the lifetime risk of heart failure [114]. The number of classical and non-classical risk factors associated with these macrovascular complications is increasing [115], some of them showing even a stronger relationship than glycemia per se [116]. In this sense, inflammation is thought to be an important mediator of this increased cardiovascular risk [117]. Thus, dietary interventions with a multifactorial approach, combining glucose-lowering, anti-inflammatory, or antioxidant properties, could be useful in the prevention of this chronic complication.

#### 4.2.1. Atherosclerosis

Four different animal models have been used to investigate the vasoprotective properties of EA (Table 3). In Apolipoprotein E-deficient mice (at a high risk of atherosclerosis but without obesity or diabetes), 3-month dietary supplementation with either pomegranate juice or pomegranate peel extracts ameliorated atherosclerotic lesion area in the aorta [118]. In a study conducted in obese Zucker rats (a model of metabolic syndrome), compared to rats fed chow, 5-week supplementation with either pomegranate juice or a pomegranate fruit extract induced significant changes in the arterial function, including decreased expression of vascular inflammation markers, increased plasma nitrate and nitrite (NOx) levels, increased endothelial NO synthase (eNOS) expression, and increased arterial vasodilatation in response to acetylcholine [119]. The latter finding was replicated in high-carbohydrate and high-fat diet-induced metabolic syndrome rats supplemented with EA for 8 weeks, along with an increase in aortic contractile response to noradrenalin [120]. Finally, in STZ-induced diabetic rats, compared to rats fed control diet, 12-week supplementation with EA reduced medial thickness enlargement, lipid deposition in the aortic arch, and vessel proliferation markers [121].

Regarding to in vitro assays, EA blunted platelet-derived growth factor-BB (PDGF-BB)-induced proliferation of primary cultures of rat aortic smooth muscle cells, by blocking PDGF receptor-β tyrosine phosphorylation, the generation of intracellular ROS, and extracellular signal-regulated kinase 1/2 (ERK1/2) downstream activation [121]. Similarly, addition of EA also corrected the impaired vasodilation of rat aortas secondary to a high-glucose environment [82]. Finally, both pomegranate juice and several pomegranate extracts decreased the intracellular cholesterol biosynthesis rate and the uptake of native LDL and oxidized-LDL in mouse peritoneal and J774A.1 macrophages, while increasing HDL-mediated cholesterol efflux capacity [118].

Such overall anti-atherosclerotic effect was not observed in humans, although clinical research on the topic is still fragmentary (Table 3). In a randomized-controlled trial conducted in adolescents with metabolic syndrome, 4-week dietary supplementation with pomegranate juice improved flow- mediated dilatation, yet at a similar level than the comparator (grape juice) [63]. In the other randomized trial performed to date, no significant differences were observed regarding flow-mediated dilatation in hypertensive patients supplemented with either pomegranate juice or water for 2 weeks [71].

#### 4.2.2. Cardiopathy

Both EA and some pomegranate extracts ameliorate several diabetic-related cardiac abnormalities in murine models (Table 3). Eight-week dietary EA reversed the cardiac effects induced by an 8-week high-carbohydrate and high-fat diet, including cardiac fibrosis, inflammatory cell infiltration, and impaired ventricular function [120]. Similarly, STZ-induced diabetic mice fed an EA-supplemented diet for 12 weeks showed lower cardiac levels of triglycerides and markers of oxidation and inflammation in comparison to mice fed chow [122]. Lower cardiac triglyceride levels and lower collagen deposit at left ventricular area and coronary artery media area were also observed in Zucker diabetic rats after a 6-week supplementation with pomegranate flower extract [123,124]. In STZ-induced diabetic rats, 3-week daily intraperitoneal injection of urolithin A or urolithin B (both resulting from the gastrointestinal metabolization of EA) reduced by approximately 30% the myocardial expression of fractalkine (a pro-inflammatory cytokine), while improving myocardium calcium transients and cardiac hemodynamic performance [106]. Finally, the cardioprotective effects of an extract of arjun tree were tested in the experimental model of myocardial infarction co-existing with diabetes. To this end, STZ-induced diabetic rats were treated with the extract for 4 weeks prior to isoproterenol-induced myocardial infarction. Forty-eight hours after the ischemic insult, rats which had been fed a diet with the extract showed lower degree of myocardial edema, inflammation, and necrosis compared to rats fed chow [30]. An in vitro assay supported the cardioprotective effect of EA observed in animals. Isolated myocardial tissue preparations from EA-treated rats showed improvements in both calcium transients and contractile function [126].

Incipient clinical research on the topic reinforced these cardioprotective properties (Table 3). In the single randomized, controlled trial conducted to date, 45 patients with coronary ischemic disease and myocardial ischemia (24% with diabetes) received either pomegranate juice or placebo drink with a similar caloric content for three months. Those patients allocated to the EA-rich food showed a decrease in stress-induced ischemia compared with the baseline, while the opposite was observed in the placebo group [66].

### 4.3. Neurologic-Related Complications of Diabetes

Even though diabetic chronic complications have been classically divided into micro- and macrovascular ones, compelling evidence suggests that T2D is also associated with other non-traditional manifestations [127]. Among them, neurological-related complications involve a high socioeconomic cost [127]. Research is needed to ascertain whether certain dietary interventions would act synergistically with glucose-lowering drugs, which have already shown promising preliminary results [128].

Five studies assessed the protective effect of EA on diabetic-related neurological manifestations. In the first one, STZ-induced diabetic rats treated with EA (50 mg/kg/day) for 3 weeks showed lower levels of oxidative stress markers in the brain or the sciatic nerve (decrease in total oxidant status, MDA, and NO) and higher activity of antioxidant enzymes when compared with rats fed chow. EA-treated animals showed lower microhemorrhagic focus with a damaged blood vessel, neuronal hydropic degenerative changes, and disorganized fibrillary degenerative changes compared to untreated rats [129]. Similar results were found in a high-fat and high-fructose diet-induced diabetic rat model, in which a 2-week treatment with EA (10 mg/kg/day) decreased brain levels of myeloperoxidase and several caspases [58]. Two studies focused on functional aspects [39,130]. In STZ-induced diabetic rats, 8-week supplementation with ground powder of pomegranate flowers at different doses (300, 400, and 500 mg/kg/day) dose-dependently ameliorated the cognitive deficits in diabetic rats (as assessed by the Morris water maze test), restoring it towards those observed in non-diabetic ones [130]. In a similar study, 8-week dietary EA (50 mg/kg/day) attenuated anxiety/depression-like behavior, improved behavioral deficits, and prevented neuronal loss [39]. Finally, administration of an extract of dried pomegranate rinds was found to improve peripheral nerve function (and the ensuing antinociceptive activity) in alloxan-induced diabetic mice [54].

## 5. Conclusions

In conclusion, experimental research fostered the notion of dietary EA as a potential agent to limit the burden of diabetes, but clinical evidence is less established. Many in vitro and animal studies explored the hypoglycemic effect of EA. In addition to improvements in both insulin sensitivity and secretion, there is evidence for the benefits of EA in decreasing postprandial hyperglycemia (via inhibition of intestinal α-glucosidase activity) and, to a lesser extent, the modulation of incretin effect. However, evidence on the favoring EA-rich dietary sources (mostly pomegranate juice) in better glucose control is much weaker in humans (Table 1). A meta-analysis reported that supplementation with pomegranate was unrelated to significant improvements in glycemic control, as assessed by fasting plasma glucose, HbA1c, insulin levels, or HOMA-IR [131]. However, trials in humans had small sample sizes, were conducted in heterogeneous populations (few including T2D individuals), and largely differed concerning the dose of pomegranate juice and the length of the intervention (often <8 weeks; ≥3 months in two studies). The most used outcome was fasting plasma glucose, while HbA1c (a variable widely used in diabetes-related studies, which integrates glucose exposure in fasting and postprandial state) was often neglected. Importantly, there are no studies using hyperinsulinemic-euglycemic clamp (the gold-standard method to assess insulin sensitivity) or studies assessing incident diabetes from prediabetic subjects. Regarding inflammation and oxidative stress, dietary EA induced significant benefits in most in vitro, animal, and human studies. Evidence on the benefits of EA in non-enzymatic glycation in humans is incipient. Experimental research found that dietary EA prevented vascular complications. However, clinical research on the topic is still scarce, with microvascular complications and diabetic neuropathy actually unexplored.

Experimental research on the topic has two main caveats. First, several in vitro studies have been conducted in cell lines (immortalized), namely HepG2 hepatocytes, 3T3-L1 pre-adipocytes, and RAW 264.7 macrophages. Therefore, observed results in these studies may not completely translate to those reported in studies involving primary cell cultures. The second one concerns the studies testing EA in isolation. It is difficult to ascertain the translation of consumption of EA-rich foods to the EA-tested doses. In this regard, several factors should be considered, including (I) the amount of EA-related compound contained in a certain type of food (i.e., the content of ellagitannins in the edible part of pomegranates), which might vary in function of many agricultural factors; (II) the coexistence with other phytochemicals in a specific food that might influence the bioavailability of EA-related compound; (III) given the key role of gut in EA metabolism and absorption, the particular microbiota profile that each subject might translate into one-to-one differences in the hydrolysis of ellagitannins and further metabolism of EA to urolithins. This reinforces the need of well-designed and correctly powered randomized controlled trials on the topic.

Large randomized controlled trials involving dietary antioxidants in isolation have failed to achieve successful results for incident diabetes [132] and cardiovascular disease [133]. A plausible explanation for these neutral findings (in apparent conflict with many observational studies) is the fact that foods rich in antioxidants may have other bioactives which are not necessarily included in tested supplements. On the other hand, while adherence to a plant-based, antioxidant-rich dietary pattern (i.e., Mediterranean Diet) has been proven to be cardioprotective [134,135], it is difficult to disentangle the exact contribution of each dietary component on a hard-clinical endpoint. Randomized controlled trials testing a single food offer an approach between single components and dietary patterns [136,137]. Therefore, given that EA improves pancreatic β-cell functionality and postprandial hyperglycemia, subjects with prediabetes / metabolic syndrome would likely obtain the highest benefit of a short-term (4 week) or mid-term (6 months) dietary supplementation with an EA-rich food (i.e., pomegranate juice). The main outcomes would be changes in meal tolerance test (to evaluate the effect on postprandial glycemia) and in intravenous glucose tolerance test (IVGTT, to evaluate pancreatic β-cell functionality). Secondary outcomes would be hyperinsulinemic-euglycemic clamp (to assess changes in insulin sensitivity) and circulating markers of inflammation/oxidative stress.

## Figures and Tables

**Table 1 antioxidants-09-01226-t001:** Human studies assessing the glucose-lowering effects of ellagic acid (EA).

Reference	Sample Size	Study Design	Participants’ Characteristics	T2D (%)	Age (Years)	Sex (M/F)	BMI (kg/m^2^)	Interventions	Duration	Main Results
*Pomegranate juice*										
[61]	85	NR, SA	T2D (85)	100	NA	40/45	NA	1.5 mL/kg bw	1 single dose	• ↓FPG and HOMA-IR
										• ↑HOMA %B after 3 h of administration
[62]	16	R, SB, C	Healthy	0	31 ± 5	NA	23 ± 3	PJ 200 mL vs. Placebo (sugar-balanced water), before bread consumption	1 single dose	• ↓incremental AUC and peak glucose after PJ
[63]	30	R, P	MetS, adolescents	0	13.4 ± 1.1	16/14	27.1 ± 1.1	PJ (240 mL/d) vs. grape juice 18 mL/kg/d	1 single dose	• ≈FPG (both arms) after 4 h of consumption
									4 wks	• ≈FPG in both arms
[64]	50	NR, SA	T2D	100	45 ± 8	NA	30 ± 3	PJ 200 mL/d	6 wks	• ↓FPG
[65]	31	NR, SA	T2D	100	46.0 ± 8.3	15/16	29.5 ± 0.7	concentrated PJ 50 g/d	4 wks	• ≈FPG
[66]	45	R, DB, P	stable CHD	24	69.0 ± 10.0	40/5	28.5 ± 5.6	PJ 240 mL/d vs. Placebo (modified sports beverage)	12 wks	• ≈FPG and HbA1c
[67]	60	R, SB	T2D	100	54.7 ± 8.4	30/30	27.3 ± 3.7	PJ 200 mL/d vs. Control (untreated)	6 wks	• ≈FPG
[68]	30	R, DB, C	MetS	0	51.6 ± 10.0	13/17	NA	PJ 500 mL/d vs. Placebo (equivalent in sugar and acid content, without polyphenols)	1 wk	• ≈FPG
										• ≈insulin
										• ≈HOMA-IR
[69]	30	R, DB, P	stable COPD	0	61.7 ± 10.0	NA	31.0 ± 5.3	PJ 400 mL/d vs. Placebo (synthetic flavored drink)	5 wks	• ≈FPG
[70]	20	R, DB, P	Obesity	0	37.3 ± 9.4	NA	34.5 ± 3.6	PJ 120 mL/d vs. Placebo	4 wks	• ≈FPG, AUC glucose
										• insulin, AUC insulin
										• insulinogenic index, Matsuda index
[71]	28	R, DB, C	Healthy	0	50.4 ± 6.1	12/16	26.8 ± 3.4	PJ 500 mL/d vs. Placebo (water plus equivalent carbohydrates)	4 wks	• ↓FPG,
										• ↓insulin and HOMA-IR
[72]	44	R, DB, P	T2D	100	55.9 ± 6.7	23/21	29.0 ± 4.0	PJ 250 mL/d vs. Placebo (equivalent in sugar content)	12 wks	• ≈FPG
										• ≈insulin or HOMA-IR
[73]	21	R, SB, P	Hypertension	0	52.9 ± 8.7	6/15	27.4 ± 3.8	PJ 150 mL/d vs. Placebo (water)	2 wks	• ≈FPG
[74]	77	R, DB, P	Overweight, women	0	41.5 ± 12.5	0/77	28.4 ± 2.2	PV-based beverage 200 mL/d vs. Placebo (equivalent in sugar and acid content)	8 wks	• ≈FPG,
										• ≈FPI, HOMA-IR
[75]	20	R, DB, P	Endurance-based athletes	0	35.3 ± 9.9	20/0	NA	PJ 200 mL/d vs. PJ diluted 1:1 with water (200 mL/d) vs. seasonal fruit (equivalent energy)	3 wks	• ≈FPG
[76]	10	R, SB	Healthy	0	31.8 ± 6.6	5/5	NA	PJ 500 mL/d vs. Placebo	2 wks	• ≈FPG
*Pomegranate extracts*										
[62]	16	R, DB, C	Healthy	0	26 ± 6	NA	23 ± 2	400 mg PE vs. 200 mg placebo+200 mg PE vs. 400 mg placebo (before bread consumption)	1 single dose	• ≈Incremental AUC
										• ≈peak glucose
[77]	64	R, DB, P	Overweight and increased waist size	0	35–65	NA	NA	PFE 710 mg or 1420 mg vs. Placebo	4 wks	• ≈FPG
[78]	42	R, DB, P	Overweight/obese	NA	30–60	NA	31.8 ± 4.5	PFE 1000 mg/d vs. Placebo (cellulose capsules)	30 d	• ↓FPG,
										• ↓FPI, HOMA-IR
*Pomegranate-based dietary supplements*										
[79]	20	NR, SB	Abdominal obesity	0	48 ± 4	10/10	29.7 ± 2.7	Dietary supplement prepared from pomegranate and grape pomaces (50:50) 10 g vs. Control (untreated)	1 single dose	• ≈Glucose after OGTT (either administered simultaneously or 10 h before of the OGTT)
										• ≈HOMA-IR

AUC: area under the curve; BMI: body mass index; bw: bodyweight; C: cross-over; CHD: coronary heart disease; COPD: chronic obstructive pulmonary disease; DB: double-blind; FPG: fasting plasma glucose; FPI: fasting plasma insulin; HOMA-IR: homeostatic model assessment of insulin resistance; HOMA %B: homeostatic model assessment of β-cell function; MetS: metabolic syndrome; NA: not available; NR: non-randomized; P: placebo-controlled; PE: pomegranate extract, PFE: pomegranate fruit extract; PJ: pomegranate juice; PV: pomegranate vinegar; R: randomized; SA: single arm; SB: single-blind; T2D: type 2 diabetes; d: days; wks: weeks. Placebo was not described for some of the studies.

**Table 2 antioxidants-09-01226-t002:** Animal studies assessing the effects of EA on microvascular diabetic complications.

Reference	Study Model	Interventions	Duration	Main Results
***Diabetic Kidney Disease***				
[41]	STZ-induced diabetic mice (BALB/c), males	Control vs. PPE-AuNP 5 (15 or 25 mg/kg every 2nd day)	10 d	• ↓renal histopathology alterations
				• ↓renal fibrosis markers (TGF-β and Col IV)
				• ↓renal oxidative stress (↓ROS, ↓LPO, ↓nitrite; ↓MAPK pathway; ↓NOX4 and p47^phox^ expression; ↑SOD, ↑GSH, ↑Nrf2 activation)
				• ↓renal inflammation (↓NF-κB and STAT3 pathways: IL-1β, IFN-γ, IL-6, IL-10, TNF-α, COX2)
				• ↓renal AGEs and RAGE
				• ↓urinary urea and creatinine
[97]	STZ-induced diabetic rats (Wistar), males	Control vs. EA (0.2 or 2%)	12 wks	• ↓protein glycation in glomeruli (IgG cross-linking, CML accumulation)
				• ↓ renal RAGE expression
				• ↓ renal TGF-β expression
				• ↓accumulation of ECM
				• ↑podocyte specific markers (podocin, nephrin)
				• ↓UACR and urinary urea
[101]	STZ-induced diabetic mice (Balb/cA), males	Control (CA, 2.5 or 5%) vs. EA (2.5 or 5%)	12 wks	• ↓renal AGEs (CML, pentosidine, sorbitol, fructose)
				• ↓renal polyol pathway (↓AR and SDH activity)
				• ↓renal inflammation markers (IL-1β, IL-6, TNF-α, MCP-1)
				• ↓serum BUN
				• ↑creatinine clearance
[102]	STZ-induced diabetic rats (Wistar), +/−Ang II), males	Control vs. PJ (100 or 300 mg/kg/d)	4 wks	• ↓renal histopathology tubular alterations (no mesangial protection)
				• ↓renal oxidative stress/lipid peroxidation (↓MDA, ↑GSH, ↑SOD, ↑CAT)
[103]	STZ-induced diabetic rats (Wistar), either sex	Control vs. PLE (50, 100 or 200 mg/kg/d)	4 wks	• ↓renal histopathology alterations and inflammatory cells infiltration
				• ↓renal oxidative stress/peroxidation (↓MDA, ↑GSH, ↑SOD, ↑CAT)
				• ↓albuminuria
				• ↓Serum BUN and creatinine
				• ↑creatinine clearance
[104]	STZ-induced diabetic mice (ICR), males	Control vs. EA (50, 100 or 150 mg/kg/d) vs. IRB (180 mg/kg/d)	4 wks	• ↓renal histopathology alterations and inflammatory cells infiltration (≈IRB)
				• ↓renal inflammation (↓TLR4/NF-κB)
				• ↓albuminuria (≈IRB)
				• ↓serum creatinine (≈IRB)
				• ↓serum oxidative stress (↓MDA and ↑SOD levels) (≈IRB)
[105]	HFD/low-dose STZ-induced type 2 diabetic rats (Wistar), males	Control vs. EA (20 or 40 mg/kg/d)	16 wks	• ↓renal histopathology tubular alterations (no mesangial protection)
				• ↓renal PAS accumulation
				• ↓renal expression of NF-κBp65, TGF-β and fibronectin
				• ↓serum inflammatory cytokines (IL-1β, IL-6, TNF-α)
				• ↓renal oxidative stress/peroxidation (↓MDA, ↑GSH, ↑antioxidant enzymes activity)
				• ↓serum creatinine, BUN, proteinuria
				• ↑creatinine clearance
***Retinopathy***				
[109]	STZ-induced diabetic rats (Wistar-NIN), males	Control vs. EA (0.2 or 2%)	12 wks	• ↑retinal thickness
				• ↓retinal AGE (CML) and RAGE
				• ↓retinal pro-apoptotic markers (Bax)
				• ↓retinal neovascularization markers (HIF-1α and VEGF)
				• ↓retinal gliosis markers (GFAP)
				• Improvement electroretinogram abnormalities
[110]	STZ-induced diabetic rats (Sprague-Dawley), either sex	Control vs. PJ (100 µL/d)	10 wks	• ↓renal histopathology tubular alterations (no mesangial protection)
				• ↓retinal oxidative stress/lipid peroxidation (↓8OHdG, ↓MDA, ↑GSH, ↑GSH-Px)
				• ↓retinal eNOS and P65 staining

8OHdG: 8-hydroxy-2′-deoxyguanosine; AGEs: advanced glycation end products; AngII: angiotensin II; AR: aldose reductase; BUN: blood urea nitrogen; EA: ellagic acid; ECM: extracellular matrix; CA: caffein acid; CAT: catalase; CML: carboxymethyllysine; Col: Collagen; d: days; GSH: glutathione; GSH-Px: glutathione peroxidase; HFD: high-fat diet; HIF-1α: hypoxia-inducible factor-1α; IL: interleukin; IL-6: interleukin-6; IFN-γ: interferon-γ; IRB: irbesartan; LPO: lipoperoxides; MCP-1: monocyte chemoattractant protein-1; MDA: malondialdehyde; NF-κB: nuclear factor-κB; PAS: periodic acid-Schiff; PLE: pomegranate leave extract; PPE-AuNP: pomegranate peel extract-stabilized gold nanoparticles; RAGE: receptor of AGE; SDH: sorbitol dehydrogenase; SOD: superoxide dismutase; STZ: streptozotocin; TGF-β: transforming growth factor-β; TNF-α: tumor necrosis factor-α; UACR: urinary albumin-to-creatinine ratio; VEGF: vascular endothelial growth factor; wks: weeks.

**Table 3 antioxidants-09-01226-t003:** Animal and human studies assessing the effects of EA on macrovascular diabetic complications.

Reference	Study Model	Interventions	Duration	Main Results
Atherosclerosis, animal studies				
[118]	ApoE-deficient mice, males	Control vs. PPE or PAE or PFE or PSE (200 µg GAE/d)	13 wks	• ↓atherosclerotic lesion size (no effect in PAE or PSE)
				• ↓serum lipoperoxides (PFE and PAE)
				• ↑serum PON1 activity (only PAE)
				• ↓macrophage oxidized-LDL uptake (only PPE)
				• ↑HDL-cholesterol efflux (only PAE)
		Control vs. PJ (200 µg GAE/d)	13 wks	• ↓atherosclerotic lesion size
				• ↓serum lipoperoxides
				• ↑serum PON1 activity
				• ↓macrophage LDL and oxidized-LDL uptake
				• ↑HDL-cholesterol efflux
[119]	Obese Zucker rats, females	Control (atherogenic diet) vs. PFE (30 µL/d)- supplemented atherogenic diet	5 wks	• ↑vasodilatation in response to acetylcholine
		Control (atherogenic diet) vs. PJ (30 µL/d)- supplemented atherogenic diet	5 wks	• ↑vasodilatation in response to acetylcholine
[120]	High-carbohydrate, HFD-induced MetS rats (Wistar), males	Control vs. EA (0.8 g/kg food)	8 wks	• ↑vasodilatation in response to acetylcholine
				• ↑aortic contractile in response to noradrenalin
[121]	STZ-induced diabetic rats (NIN-Wistar), males	Control vs. EA (2%)	12 wks	• ↓medial layer thickness
				• ↓lipid accumulation
				• ↓deposition of collagen
				• ↓cyclin D1 expression in media (SMC proliferation marker)
Atherosclerosis, human studies				
[63]	MetS, adolescents (n = 30)	PJ (240 mL/d) vs. grape juice (18 mL/kg/d)	4 wks	• ↑FMD, both interventions
[73]	Hypertension (n = 20)	Placebo (water) vs. PJ (150 mL/d)	2 wks	• no improvements in FMD
Cardiopathy, animal studies				
[30]	Myocardial infarction in STZ-induced diabetic rats (Wistar), males	Control vs. *Terminalia arjuna* extract (500 mg/kg/d) or vildagliptin (10 mg/kg/d)	4 wks	• ↓focal myofiber loss
				• ↓inflammation
				• ↓necrosis
				• ↓edema
				• ↓cardiac parameters (CPK-MB, but less than vildagliptin)
[120]	High-carbohydrate, HFD-induced MetS rats (Wistar), males	Control vs. EA (0.8 g/kg food)	8 wks	• ↓cardiac infiltration of inflammatory cells
				• ↓cardiac collagen deposition
				• ↑cardiac hemodynamic performance (↑EF; ↓LVIDs; ↓systolic volume, ↓fractional shortening; ↓estimated LV mass; ↓LV diastolic stiffness)
				• ↓cardiac NF-κB expression
				• ↑cardiac CPT-1 expression
[122]	STZ-induced diabetic mice (Balb/c), males	Control vs. EA (2%)	12 wks	• ↓cardiac triglyceride content (but not cholesterol)
				• ↓cardiac inflammation (IL-1β; IL-6; TNF-α; MCP-1)
				• ↓cardiac oxidative stress (↓MDA, ↓ROS; ↑GSH, ↑CAT, ↑SOD, ↑GSH-Px)
[123]	Zucker diabetic fatty rats, males	Control vs. PFE (500 mg/kg/d)	6 wks	• ↓cardiac triglyceride content (but not cholesterol)
				• ↓abnormal cardiac upregulation lipogenic genes (FATP, PPARα, CPT-1, ACO, AMPKα2)
[124]	Zucker diabetic fatty rats, males	Control vs. PFE (500 mg/kg/d)	6 wks	• ↓interstitial and perivascular collagen deposit
				• ↓cardiac fibrosis markers (fibronectin, collagen I and III, ET-1 and ET_A_)
				• ↓NF-κB pathway activation
[125]	STZ-induced diabetic rats (Wistar), males	Control vs. Urolithin A or Urolithin B (2.5 mg/kg/d)	3 wks	• ↑cardiac hemodynamic performance, specially urolithin B (↑LVSP; ↑+dP/dt_max_, ↓IVCT, ↓Tcycle)
				• Improvement of cardiomyocyte mechanics and calcium transients
				• ↓fractalkine (cardiac pro-inflammatory cytokine)
Cardiopathy, human studies				
[66]	CHD and myocardial ischemia (n = 45)	Placebo vs. PJ (240 mL/d)	13 wks	• ↓stress-induced ischemia (in myocardial perfusion single-photon emission computed tomographic technetium-99m tetrofosmin scintigraphy)

ACO: acyl-CoA oxidase; AMPK: 5′-AMP-activated protein kinase; ApoE: apolipoprotein E; CAT: catalase; dP/dtmax: maximal rate of ventricular pressure rise; CHD: coronary heart disease; CPK-MB: creatinine phosphokinase-MB; CPT-1: carnitine palmitoyltransferase 1; EA: ellagic acid; EF: ejection fraction; ET: endothelin; FATP: fatty acid transport protein; FMD: flow-mediated dilatation; GAE: gallic acid equivalents; GSH; glutathione; GSH-Px: glutathione peroxidase; HDL: high-density lipoprotein; HFD: high-fat diet; IL: interleukin; IVCT: isovolumic contraction time; LDL: low-density lipoprotein; LV: left ventricular; LVIDs: left ventricular internal diameter during systole; LVSP: left ventricular systolic pressure; MCP-1: monocyte chemoattractant protein-1; MDA: malondialdehyde; MetS: metabolic syndrome; NF-κB: nuclear factor-kB; PAE: pomegranate arils extract; PFE: pomegranate flowers extract; PJ: pomegranate juice; PON1: paraoxonase 1; PPARα: peroxisome proliferator-activated receptor-α; PPE: pomegranate peel extract; PSE: pomegranate seed extract; ROS: reactive oxygen species; SMC: smooth muscle cell; SOD: superoxide dismutase; Tcycle: total cycle duration; TNF-α: tumor necrosis factor-α; d: days; wks: weeks.

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
