# Peer review of "Ellagic Acid as a Tool to Limit the Diabetes Burden: Updated Evidence"

_antioxidants, 2020, doi:10.3390/antiox9121226_

Round 1

Reviewer 1 Report

This is a very nice and comprehensive review.  It is well organized as it utilizes the diabetic, oxidative stress/ inflammation, and complications categories to discuss the impact of ellagic acid (or foods/ extracts rich in ellagic acid) to discuss the in vitro, animal, and human evidence of its impact.  It does a very nice job of providing significant detail in the narrative, as well as the extensive table.  It also does not shy away from pointing out where the evidence is not well demonstrated.  It is well written and should be of value for scientists in the field.  It may be helpful to point out when discussing cell culture work that transformed cells are "cancer" cells and the results may not translate to healthy cells.  Also, when discussing cell culture and animal studies it would be helpful to remind readers how he given doses translate to food sources of ellagic acid.

Author Response

Thank you for your thorough review and your comments, which helped us to improve the manuscript. Regarding the first comment, the reviewer is right on that because of the cancer-like behaviour, results from studies involving immortalized cells may not be reproducible to those involving healthy/primary cells. Regarding the second part of the critique, it is difficult to ascertain the real extent to which EA contained in a food reaches the cell as EA, since many factors should be considered. At the end of the “Conclusions” of the revised version of the manuscript, we included both notions as main limitations of the experimental research on the topic. In line 563-576, please find:

“Experimental research on the topic has two main caveats. First, several in vitro studies have been conducted in cell lines (immortalized), namely HepG2 hepatocytes, 3T3-L1 pre-adipocytes, and RAW 264.7 macrophages. Therefore, observed results in these studies may not completely translate to those reported in studies involving primary cell cultures. The second concerns the studies testing EA in isolation. It is difficult to ascertain the translation of consumption of EA-rich foods to the EA tested doses. In this regard, several factors should be considered, including (i) the amount of EA-related compound contained in a certain type of food (i.e., the content of ellagitannins in the edible part of pomegranates), which might vary in function of many agricultural factors; (ii) the coexistence with other phytochemicals in a specific food that might influence the bioavailability of EA-related compound; (iii) given the key role of gut in EA metabolism and absorption, the particular microbiota profile that each subject might translate into one-to-one differences in the hydrolysis of ellagitannins and further metabolism of EA to urolithins. This reinforces the need of randomized controlled trials on foods rich in EA-related compounds”.

Reviewer 2 Report

This review is generally well written but several issues should be considered in relation to clinical translation:

  1. The effect of nutrition on health in general depends on the interaction between components of a specific dietary pattern. It is questionable that one specific compound may result in a effect on a hard clinical endpoint.
  2. Interventions with antioxidants in previous randomized trials have been disappointing.
  3. When considering an intervention in humans, the authors should clarify whether they intend to randomize a purified compound or specific nutrients that contain the compound of interest.
  4. The authors should also clarify which specific hard clinical endpoint would be considered in such trial.
  5. There is one example of successful intervention in humans with a pure compound: the REDUCE-IT trial, which demonstrated a powerful effect of icosapent ethyl (Vascepa). Many subjects in this trial had diabetes. On the other hand, examples of the effect of a dietary pattern are the Lyon Diet Heart study and the PREDIMED trials.
  6. The authors rightly conclude that the (circumstantial) evidence presented does not constitute a proof. In light of this, they should be specific on the design of a clinical trial to refute or confirm a specific research hypothesis

Author Response

Thank you for your comments, which highlights your knowledge in the field. We thoroughly considered them, and we thank you for helping us in improving the final “Discussion”, in particular concerning to the translation into human research. In line 577-592, please find:

“Large randomized controlled trials involving dietary antioxidants in isolation have failed to achieve successful results for incident diabetes [132] and cardiovascular disease [133]. A plausible explanation for these neutral findings (in apparent conflict with many observational studies) is the fact that foods rich in antioxidants may have other bioactives which are not necessarily included in tested supplements. On the other hand, while adherence to a plant-based, antioxidant-rich dietary pattern (i.e., Mediterranean Diet) has been proven to be cardioprotective [134, 135], it is difficult to disentangle the exact contribution of each dietary component on a hard-clinical endpoint. Randomized controlled trials testing a single food offer an approach between single components and dietary patterns [136, 137]. Therefore, given that EA improves pancreatic β-cell functionality and postprandial hyperglycemia, subjects with preadiabetes / metabolic syndrome would likely obtain the highest benefit of a short-term (4 week) or mid-term (6 months) dietary supplementation with an EA-rich food (i.e., pomegranate juice). The main outcomes would be changes in meal tolerance test (to evaluate the effect on postprandial glycemia) and in intravenous glucose tolerance test (IVGTT, to evaluate pancreatic β-cell functionality). Secondary outcomes would be hyperinsulinemic-euglycemic clamp (to assess changes in insulin sensitivity) and circulating markers of inflammation / oxidative stress”.

Please also find the newly added references in the revised version:

  1. Kataja-Tuomola M, Sundell JR, Männistö S, Virtanen MJ, Kontto J, Albanes D, Virtamo J. Effect of alpha-tocopherol and beta-carotene supplementation on the incidence of type 2 diabetes. Diabetologia. 2008 Jan;51(1):47-53. doi: 10.1007/s00125-007-0864-0.
  2. Jenkins DJA, Spence JD, Giovannucci EL, Kim YI, Josse R, Vieth R, Blanco Mejia S, Viguiliouk E, Nishi S, Sahye-Pudaruth S, Paquette M, Patel D, Mitchell S, Kavanagh M, Tsirakis T, Bachiri L, Maran A, Umatheva N, McKay T, Trinidad G, Bernstein D, Chowdhury A, Correa-Betanzo J, Del Principe G, Hajizadeh A, Jayaraman R, Jenkins A, Jenkins W, Kalaichandran R, Kirupaharan G, Manisekaran P, Qutta T, Shahid R, Silver A, Villegas C, White J, Kendall CWC, Pichika SC, Sievenpiper JL. Supplemental Vitamins and Minerals for CVD Prevention and Treatment. J Am Coll Cardiol. 2018 Jun 5;71(22):2570-2584. doi: 10.1016/j.jacc.2018.04.020.
  3. de Lorgeril M, Salen P, Martin JL, Monjaud I, Delaye J, Mamelle N. Mediterranean diet, traditional risk factors, and the rate of cardiovascular complications after myocardial infarction: final report of the Lyon Diet Heart Study. Circulation. 1999 Feb 16;99(6):779-85. doi: 10.1161/01.cir.99.6.779.
  4. Estruch R, Ros E, Salas-Salvadó J, Covas MI, Corella D, Arós F, Gómez-Gracia E, Ruiz-Gutiérrez V, Fiol M, Lapetra J, Lamuela-Raventos RM, Serra-Majem L, Pintó X, Basora J, Muñoz MA, Sorlí JV, Martínez JA, Fitó M, Gea A, Hernán MA, Martínez-González MA; PREDIMED Study Investigators. Primary Prevention of Cardiovascular Disease with a Mediterranean Diet Supplemented with Extra-Virgin Olive Oil or Nuts. N Engl J Med. 2018 Jun 21;378(25):e34. doi: 10.1056/NEJMoa1800389.
  5. Balfegó M, Canivell S, Hanzu FA, Sala-Vila A, Martínez-Medina M, Murillo S, Mur T, Ruano EG, Linares F, Porras N, Valladares S, Fontalba M, Roura E, Novials A, Hernández C, Aranda G, Sisó-Almirall A, Rojo-Martínez G, Simó R, Gomis R. Effects of sardine-enriched diet on metabolic control, inflammation and gut microbiota in drug-naïve patients with type 2 diabetes: a pilot randomized trial. Lipids Health Dis. 2016 Apr 18;15:78. doi: 10.1186/s12944-016-0245-0.
  6. Sala-Vila A, Valls-Pedret C, Rajaram S, Coll-Padrós N, Cofán M, Serra-Mir M, Pérez-Heras AM, Roth I, Freitas-Simoes TM, Doménech M, Calvo C, López-Illamola A, Bitok E, Buxton NK, Huey L, Arechiga A, Oda K, Lee GJ, Corella D, Vaqué-Alcázar L, Sala-Llonch R, Bartrés-Faz D, Sabaté J, Ros E. Effect of a 2-year diet intervention with walnuts on cognitive decline. The Walnuts And Healthy Aging (WAHA) study: a randomized controlled trial. Am J Clin Nutr. 2020 Mar 1;111(3):590-600. doi: 10.1093/ajcn/nqz328.

Round 2

Reviewer 2 Report

The authors have provided an adequate response to the comments of the reviewer.